# Molecular Fingerprinting and Microbiological Characterisation of Selected *Vitis vinifera* L. Varieties

**DOI:** 10.3390/plants11233375

**Published:** 2022-12-05

**Authors:** Jozef Sabo, Silvia Farkasová, Michal Droppa, Jana Žiarovská, Miroslava Kačániová

**Affiliations:** 1Institute of Horticulture, Faculty of Horticulture and Landscape Engineering, Slovak University of Agriculture, Tr. A. Hlinku 2, 94976 Nitra, Slovakia; 2Institute of Plant and Environmental Sciences, Faculty of Agrobiology and Food Resources, Slovak University of Agriculture in Nitra, Tr. A. Hlinku 2, 94976 Nitra, Slovakia

**Keywords:** *Vitis vinifera* L, varietal variability, CDDP fingerprinting, chitinase expression, antimicrobial activity

## Abstract

The aim of the study was to analyse selected aspects of the natural variability of selected varieties of *Vitis vinifera*. Grapevine is one of the most popular and desirable crops in the world due to the great tradition of wine production, but grape extracts also have a wide range of pharmaceutical effects on the human body. It is important to identify different varieties for the conservation of genetic resources, but also for commercial and cultivation purposes. The variability of conserved DNA-derived polymorphism profiles, as well as microbial characteristics, were analysed in this study. Six different varieties of *Vitis vinifera* L. were used in the study: Cabernet Savignon, Chardonney, Welschriesling, Weisser Riesling, Gewurztramines and Gruner Veltliner. Genetic polymorphism was analysed by CDDP markers for *WRKY* genes. Polymorphic amplicon profiles were generated by all primer combinations used in the study. Gruner Veltliner and Welschriesling were the most similar, with a similarity value at 0.778. Microbiological quality of grape and antimicrobial activity against Gram-positive and Gram-negative bacteria and yeasts were analysed further. The plate diluting method for microbial quality and the disc diffusion method for antimicrobial activity were evaluated. The number of total count of bacteria ranged between 3.12 in Cabernet Sauvignon to 3.62 log cfu/g in Grűner Veltliner. The best antimicrobial activity showed Gewurztramines against *Salmonella enterica*, *Yersinia enterocolitica*, *Pseudomonas aeroginosa*, *Staphylococcus aureus*, *Listeria monocytogenes*, *Candida albicans*, *Candida krusei*, and *Candida tropicalis*. The best antimicrobial activity was found against *Enterococcus faecalis* in variety Weisser Riesling.

## 1. Introduction

The grapevine (*Vitis vinifera* L.) is native to southern Europe and Western Asia and is cultivated nowadays in all temperature regions of the world [1]. *Vitis vinifera* L. is a vegetatively propagated plant and is one of the oldest cultivated plants. Grapes are used in many forms in food production as well as in pharmacy. More than 6000 varieties have been recognised based on their morphological character. Since the grapevine has vital economical value, the viticulture industry has interest in the identification of the different vegetatively propagated lines [2].

The cultivation of the grapevine worldwide is connected to different biotic stresses that this plant needs to be able to interact with. Different mechanisms are involved, such as transcriptional activation of defence-related genes, deposition of mechanical barriers, accumulation of phytoalexins and pathogenesis-related proteins synthesis [3]. Many types of pathogen-related proteins are produced by the plant, among them chitinases, which are among the best characterised pathogen-related proteins. Their physiological functions were reported in a wide range of different roles in embryogenesis and ethylene synthesis in response to environmental stresses [4], or in direct response to phytopathogen attack [5,6]. Defence proteins are parts of the stems, flowers and seeds a well as tubers of plants [6]. Four different types of chitinases are characterised in plants [3,7,8,9,10,11]. Chitinases of class I, class III and class IV were reported in *Vitis vinifera* L. [3,8,9]. Class IV of chitinases is connected with the resistance of fungal diseases [10,12], but alongside the high expression of this class under the biotic attack, in grape berries, class I was reported to be overexpressed, too [3].

Antibacterial characteristics of plants are specific one among the defence mechanisms against biotic attacks. Only a few chitinases are reported as proteins with antimicrobial effect [13]. Class I chitinase is naturally more antifungal when compared to other classes [14]. Chitinase activity in berry skin extracts was confirmed previously [15].

Antimicrobial compounds of plants are mainly polyphenols. Grapes are considered a rich source of these molecules as 60–70% of extractable polyphenols in grapes are located in the seeds. The antimicrobial activity of different kinds of grape extracts were investigated previously against various bacterial species. GPEs were proved to be effective against Gram-positive and Gram-negative bacteria and yeasts [16]. The antibacterial and antifungal activities of the grape extracts were evaluated using four strains of bacteria (*Staphylococcus aureus*, *Bacillus cereus*, *Escherichia coli* and *Pseudomonas aeruginosa*) and three fungi strains (*Candida albicans*, *Candida parapsilosis*, *Candida krusei*) and, in fluid extracts, the highest antimicrobial effectiveness was obtained compared to the other grape pomace extracts due to the presence of antimicrobial active compounds [17].

Molecular markers have been demonstrated to be a competent device for fingerprinting, assessing genetic variation and studying the relatedness of cultivars of numerous species. This is due, in part, to their not being affected by the environment [18]. Nowadays, many new molecular marker techniques are known in plant genetics.

Conserved DNA or gene family-based markers are a special group of gene-targeted markers (GTMs) which utilise length polymorphisms of exon-intron structures in different widely distributed and common plant genes or gene families. Conserved genes, or, ideally, sequences of gene families, are present in multiple copies in the plant genome [19]. Conserved DNA regions are observed within a selection of well characterised plant genes involved in plant development and the development of DNA markers in response to abiotic and biotic stresses [20]. These short, conserved gene sequences occur at multiple sites within plant genomes and, therefore, provide multiple primer binding sites. In our study, we used the primers for conserved DNA sequence targeting of WRKY genes. The group of WRKY proteins are a superfamily of regulators that control diverse developmental and physiological processes. The WRKY family was first described to be plant specific, but later the identification of WRKY genes in nonphotosynthetic eukaryotes showed them to have conserved DNA sequences [20].

The CDDP technique is based on the single primer amplified region principle and uses a single primer as a forward and reverse primer. This technique was proved to be universal in its use with good discrimination potential in individual plant species with the polymorphism around 90%. CDDP markers have been used by researchers to reveal genetic polymorphism in many plant species, for example, in *Triticum* species, *Hordeum vulgare*, *Thinopyrum* species [21], chickpea [22], rose (*Rosa* spp.) [23], peony (*Paeonia suffruticosa*) [24], chrysanthemum [25], *Salix taishanensis* [26] and others.

The aim of this study was to characterise the natural variability of microbial and antimicrobial characteristics of selected varieties of *Vitis vinifera* L., their coding parts DNA polymorphism and the expression of class I chitinase differences among matured and unmatured grapes.

## 2. Results

### 2.1. Microbiological Characterisation

The total count of bacteria ranged between 3.19 ± 0.06 in Cabernet Sauvignon to 3.51 ± 0.10 log cfu/g in Grűner Veltliner. The number of coliform bacteria ranged between 1.19 ± 0.04 in Cabernet Sauvignon to 1.47 ± 0.03 in Weisser Riesling and 1.47 ± 0.07 log cfu/g in Welschriesling. Microscopic filamentous fungi ranged between 1.14 ± 0.02 in Cabernet Sauvignon and 1.14 ± 0.05 in Chardonnay to 1.21 ± 0.06 log cfu/g in Chardonnay (Table 1). Total bacterial counts on different wine grape berries ranged from 2.57 ± 0.09 in Chardonnay to 4.39 ± 0.21 log CFU·g^−1^ in Pálava.

In the group of total viable count, significant difference was found between Cabernet Sauvignon and Grűner Veltliner (Table 1, Figure 1). For coliform bacteria, two groups of grapevine varieties were detected where differences were obtained. For microscopic filamentous fungi, no differences were found among analysed varieties of *Vitis vinifera* L.

Using the same matured grapes, antimicrobial activity was investigated further. The diameters of the inhibition zones (in mm) that correspond to the tested pomace extracts are shown in Table 2.

The best antimicrobial activity showed Gewurztraminer against *S. enterica* (22.3 ± 0.57 mm), *Y. enterocolitica* (21.6 ± 0.57 mm), *P. aeroginosa* (19.3 ± 0.57 mm), *S. aureus* (16.3 ± 0.57 mm), *L. monocytogenes* (16.6 ± 0.57 mm), *C. albicans* (11.3 ± 0.57 mm), *C. krusei* (10.6 ± 0.57 mm) and *C. tropicalis* (9.3 ± 0.57 mm). The best antimicrobial activity was found against *E. faecalis* (14.6 ± 0.57 mm) in the Weisser Riesling variety (Table 2).

In all of the analysed species tested for antimicrobial activity, statistical differences were observed among tested grapevine varieties. For Pseudomonas aerogonosa, only varieties Chardonnay and Gewurztraminer were of different antimicrobial effect. For all of the others microbial species, differences in analysed grapevine varieties were more visible (Table 2).

The obtained results suggest that the grape pomaceae extracts are effective against bacteria and yeasts. When regarding individual varieties, different antimicrobial activity was obtained for individual tested microorganisms, with significant differences mainly among bacteria and yeasts data.

### 2.2. Conserved DNA-Derived Polymorphism Fingerprints

Using the same samples as those used for microbial and antimicrobial characteristics, molecular analysis was investigated further. The conserved DNA-derived polymorphism (CDDP) marker techniques were used to analyse genetic polymorphism in 10 rice varieties [20]. In the next section, we report the results of the examination of genetic polymorphism in six varieties of grapevine. WRKY primers were used for the fingerprints amplification in analysed grapevine varieties. The combination of five primers gave us 101 fragments in total. The variety with the highest number of amplicons was Weisser Riesling and the one with the lowest number of amplicons was the Gewurztraminer variety.

The primer pair WRKY R1 and F1 produced 38 fragments, the fragments being between 65 and 2553 base pairs (bp) in size. None of the analysed variety generated the same band pattern. Grűner Veltliner and Weisser Riesling had the most similar profile of fragments when using primers WRKY R2 and WRKY F1; the length of the fragments was between 85 and 2493 bp and the primer pair gave us 11 amplicons. In contrast, the Chardonay and Cabernet sauvignon varieties had the most unsimilar amplicon profiles using primers WRKY R2B and F1, with 17 fragments of 17 to 1629 bp in size. The primer pair WRKY R3 and F1 produced 19 fragments, all studied varieties had a specific amplicon pattern (Figure 2). WRKY R3B and F1 primer pair produced 16 amplicons (49 bp–1978 bp) with less differences among the observed varieties. Amplicons at size 91 bp, 189 bp and 278 bp occurred in all varieties.

Constructed dendograms (Figure 3) were based on Jaccard genetic coefficients of similarity and UPGMA algorithms.

The conserved DNA-derived polymorphism method proved to be effective for grapevines with all five primer pairs (Table 3). High diversity was proved among the studied varieties and the identification of the separate genotypes was demonstrated. For all of the used primer combinations, a complete distinguishing of analysed varieties was obtained, and the level of polymorphism ranged from 90% to the 97%. A repetitive grouping pattern of Grűner Veltliner and Weisser Riesling and of Welschriesling and Chardonnay was obtained in some of the CDDP primer combinations.

The following statements are based on our practical findings: the presence of monomorphic profile was not found in the case of the use of all five primer pairs in all six observed varieties; the values of Jaccard coefficients of genetic similarity were in the range from 0.778 to 0.143; Weisser Riesling and Grűner Veltliner had the most similar amplicon profiles with the use of primers WRKY R1/F1 and WRKY R2/F1.

In our study, the degree of polymorphism in each variety with all CDDP WRKY based primer combinations was sufficient, and we did not observe a complete match in the amplicon profiles.

### 2.3. Chitinase Expression

Expression profiles of Class I chitinase active in the defence mechanism against pathogens were compared in analysed varieties of unmatured and matured grapes. Based on the CDDP fingerprint analysis, Gewurztraminer was selected as calibrator, being the most distinctive in its profile. Expression changes were calculated by delta–delta Ct method. Chitinase expression analysis showed that it was highest in the Grűner Veltliner for both unmatured (6.3 times higher as in calibrator) and matured grapes (4.6 times higher as in calibrator) (Figure 4).

For the correction of sample-to-sample variation, normalisation against the actine as the housekeeping gene was used. Dissociation curves in all samples were calculated by melting temperature analysis to be 86.2 °C for chitinase to check the specifity of amplicon generated in the real-time PCR. Ct values varied from 25.61 (Chardonney) to 33.95 (Grűner Veltliner) for unmatured grapes and from 29.42 (Gewurztraminer) to 32.5 (Grűner Veltliner) for matured grapes.

The most stable expression profiles were obtained for Weisser Riesling and Welschriesling when comparing unmatured and matured grapes. The class I chitinase expession in Cabernet Sauvignon was the most different among all the analysed varieties as, for unmatured grapes, the expression was 0.37 times higher than the calibrator, but for matured grapes, the expression was 2.9 times higher.

## 3. Discussion

Wine production is a complex process from the vineyard to the winery. On this journey, microbes play a decisive role. The grapevine (*Vitis vinifera* L.) phyllosphere harbors diverse microbes including yeasts, filamentous fungi and bacteria that substantially modulate grapevine health and growth, as well as grape and wine production [27,28]. Microbes could originate from the vineyard soil [29,30], air, precipitation (rainfall, hail, snow), be transported by animal vectors (bees, insects, and birds) [31,32,33,34] or be resident in nearby native forests [30,35]. The microscopic filamentous fungi count ranged from 1.18 ± 0.03 in Blue Frankish to 2.60 ± 0.17 log CFU.g^−1^ in Welschriesling [36].

A number of studies have been carried out on the antibacterial effect of grape pomace extracts against foodborne pathogens [37]. In addition, polyphenols of red grape pomace extract potentiated the effects of various classes of antibiotics against *Staphylococcus aureus* and *Escherichia coli*, demonstrating itself to be particularly useful to control the growth of multi-drug resistant clinical isolates [38,39].

Gram-positive bacteria were more sensitive to grape pomace treatment than Gram-negative bacteria, opposite to our study. These differences could be explained by the presence of the lipopolysaccharide cell wall in Gram-negative bacteria, which can limit the penetration of polyphenols into the cell [40,41].

Pomace extracts in different studies showed a good antimicrobial activity against *Staphylococcus aureus* and *Enterococcus faecalis* and *Pseudomonas* spp. [37]. The pomace extracts were tested to establish the effects on Gram-positive and Gram-negative bacteria. The analysed samples exhibited insignificant antibacterial activity and the method requires optimisation [42].

Genetic polymorphism varies among species and has important implications for the evolution and conservation of species [43]. The CDDP marker system has many advantages such as accessibility and significant polymorphism which can effectively provide markers related to target traits [20]. CDDP fingerprints are reported to be of good discrimination potential in individual plant species with the polymorphism around 90% and relatively high coefficients of genetic similarity, such as from 0.617 to 0.944 for chrysanthemum [25]. This is based on the highly conserved WRKY domain that is a core motif of the primer set for CDDP [20].

Previously, the CDDP technique was used to examine the genetic relationships between twenty-nine individual grape samples from Taif governorate [44]. The average polymorphism information content was at the level 0.33 and average band informativeness was 33.8. CDDP markers were summarised to be a consistent technique for the evaluation of genetic diversity and correlations among grapevine species. CDDP marker assays have successfully detected higher numbers of genetic variation between control and micropropagated samples of grapevines [45].

Plant chitinases were found in many crop species such as *Carica papaya* [14] or *Arachis hypogaea* [46]. The activity of grape chitinases was detected in various grapevine tissues and was reported to be about ten times higher in berries than in leaves [7]; however, the data concerning their expression in grape berries are still limited and mostly connected to special treatments in postharvest manipulation [3] or under direct infections [11]. Actually, only some studies are available in the literature for grape class I chitinase expression. Chitinase I was reported to be strongly expressed in the roots and stem-internodes, less expressed in berries and absent of expression in the leaves [7]. In another study, this type of chitinase was reported as unexpressed or only weakly expressed in grape varieties Muscat Gordo Blanco, Shiraz, Pinot Noir, Riesling, Semillon, Sultana, Chardonnay and Cabernet Savignon by Northern Blots of total RNA [9], which is contrary to our results, and can be a consequence of the detection limits of the methods used that differ in these characteristics. The most recent results of grape class I chitinase in the Cardinal variety under the CO_2_ pre-treatment show its variable expression in the skin tissues of non-treated grape berries and the accumulation of Vcchit1b transcript during low temperature storage was paralleled by the change in total decay [3].

## 4. Materials and Methods

### 4.1. Biological Material

Unmatured and matured grapes from six different *Vitis vinifera* L. cultivars were used in the study. They were obtained from the Sabo vineyard that belongs to the Small Carpathians wine region of Slovakia. All the samples were immediately after harvesting transported to the laboratory and kept in −50 °C until further processing. One red variety, Cabernet Sauvignon, and five white varieties, namely Chardonney, Welschriesling, Weisser Riesling, Grűner Veltliner and Gewurztraminer, were used in the analysis. All of the analysed grape varieties were planted under the same soil, climatic and stress condition during their growth.

### 4.2. Microbiological Analyses

Dilution plating (surface spread method) was used for quantitative microbiological analyses of the total viable count of microorganisms, coliform bacteria, and microscopic filamentous fungi in grape samples. Five grams of each grape sample were homogenised for 30 min in 45 mL of physiological solution (0.89%). Serial decimal dilutions up to 10^−3^ were made and aliquot of each dilution were inoculated in triplicate onto Petri dishes with selective media. The Plate Count Agar (PCA, Oxoid, Basingstoke, UK) medium for total plate count of microorganisms at 30 °C for 48–72 h, Violet red bile lactose agar (VRBL, Oxoid, Basingstoke, UK) for coliform bacteria at 37 °C for 24–48 h and Malt Extract agar (MEA, Oxoid, Basingstoke, UK) for microscopic filamentous fungi at 25 °C for 5 days were used. The dishes were kept in a thermostat in aerobic conditions. After incubation, the number of microorganisms in 1 g of the sample was calculated and expressed in log CFU/g.

### 4.3. Grape Pomace Extract Preparation

Immediately freeze-dried on receipt, the grape pomace extracts were prepared from a portion of the grape pomace samples (100 g). The samples were extracted with 96% ethanol at 1:10 ratio (*m*/*V*) under overnight shaking. The extracts were filtered through Whatman No. 2 filter paper to remove unwanted residues. After evaporating off the organic solvent, the filtrates were dissolved in dimethyl sulfoxide (0.1% DMSO) at 40 mg/mL as the stock solution and stored at −20 °C for further investigation.

### 4.4. Microorganisms Tested

In our study, nine strains of microorganisms including three Gram-negative bacteria G^−^(*Pseudomonas aeruginosa* CCM 1959, *Salmonella enteritidis* subsp. *enteritidis* and *Yersinia enterocolitica* CCM 5671), three Gram-positive bacteria G^+^ (*Staphylococcus aureus* subsp. *aureus* CCM 2461, *Enterococcus faecalis* CCM 4224 and *Listeria monocytogenes* CCM 4699) and the three yeasts strains (*Candida albicans* CCM 8186, *Candida krusei* CCM 8271 and *Candida tropicalis* CCM 8223) were tested. All tested microorganisms were collected from the Czech Collection of microorganisms (Brno, Czech Republic). The bacterial suspensions were cultured in the Mueller Hinton broth (MHB, Oxoid, Basingstoke, UK) at 37 °C for 24 h and yeasts were cultured in the Sabouraud dextrose broth (SDB, Oxoid, Basingstoke, UK) at 25 °C for 24 h.

### 4.5. Disc Diffusion Method

The agar disc diffusion method was used for the determination of the antimicrobial activity of the grape pomace extracts. Briefly, a suspension of the tested microorganism (0.1 mL of 10^5^ cells/mL) was spread onto Mueller Hinton Agar (MHA, Oxoid, Basingstoke, UK) and Sabouraud dextrose agar (Oxoid, Basingstoke, UK) at 25 °C. Filter paper discs (6 mm in diameter) were impregnated with 15 µL of the grape pomace extract and placed on the inoculated plates. Chloramphenicol (10 μg/disk, Oxoid, Basingstoke, UK) for G^−^, streptomycin (10 μg/disk, Oxoid, Basingstoke, UK) for G ^+^ and Fluconazole (25 µg/disc, Oxoid, Basingstoke, UK) for yeasts were used as a positive control to determine the sensitivity of the microorganisms under study. The plates were kept at 4 °C for 2 h and subsequently incubated aerobically at 37 °C for 24 h and 25 °C for 24 h for bacteria and yeast, respectively. The diameters of the inhibition zones were measured in millimetres. All the tests were performed in triplicate.

### 4.6. Nucleic Acid Extraction and Amplification Conditions

The total DNA extraction was performed with GeneJET Plant Genomic DNA Purification Kit (Thermo Scientific). The quality and quantity of the DNA was measured spectrophotometrically by Nanophotomoter P360 (Implen). We performed the PCR reaction in a TProfessional Basic gradient XL (Biometra) thermal cycler. We used 5 primer combinations according to [20]: WRKY-F1-WRKY-R1, WRKY-F1-WRKY-R2, WRKYF1-WRKY-R2B, WRKY-F1-WRKY-R3, WRKY-F1-WRKY-R3B (Table 4).

The reaction conditions were the following: initial denaturation at 95°C for 15 min; than 40 cycles at 95 °C for 45 s, 54 °C for 45 s, 72 °C for 90 s, and for the final extension, 72 °C for 10 min. We evaluated the reaction in 1.5% agarose gels made with 1x TBE (Tris-borate-EDTA) buffer and the DNA fragments were stained using GelRed^®^ Nucleic Acid Gel Stain (Biotium, Fremont, CA, USA). For visualisation, we used BDAdigital system 30 (Analytik Jena, Jena, Germany).

RNA was extracted using the GeneJet Plant RNA Purification Mini Kit (ThermoFisher, Waltham, MA, USA). RNA concentration and A260/A280 nm ratios were determined by Implen Nanophotometer and the integrity of the RNA was checked in 1% agarose gels. Reverse transcription and cDNA synthesis was performed from 40 ng of extracted total RNA using the Tetro cDNA synthesis kit (BIOLINE, Cincinnati, OH, USA) and oligodT primer.

A two-step real-time PCR protocol was applied for expression analysis. Actin (GenBank accession AY847627) was used as housekeeping gene. 5x Hot FirePol EvaGreen (Solis BioDyne, Tartu, Estonia) was used in amplification and the analysis was performed by in Stratagene Mx3005P (Agilent, Santa Clara, CA, USA) under the following reaction conditions: initial denaturation at 95 °C for 2 min, then 40 cycles of 95 °C for 10 s and 60 °C for 40 s followed by final analysis of amplicon dissociation curves. Class I chitinase specific primers were used.

### 4.7. Data Analysis

Two factor ANOVA with replications were prepared and performed in Microsoft Excel for Windows.

The separated CDDP amplicons were transformed to binary matrices based on the presence or absence of the band. Based on binary matrixes, UPGMA dendrograms were made by online tool http://genomes.urv.cat/UPGMA/ (accessed on 26 August 2022).

A qPCR analysis with the biological triplicates was used in the study and the relative expression values were calculated by the delta–delta Ct method [47] when the expression of chitinase was determined as the number of amplification cycles obtained in the reaching of the threshold during the exponential phase of the PCR.

## 5. Conclusions

The best antimicrobial activity from the varieties was found in Gewurztraminer against eight of the nine microorganisms. The numbers of total viable count, coliform bacteria and microscopic fungi were very balanced. The efficiency of the CDDP method in terms of determining diversity in grapevine genotypes was examined in our study. We can state that the CDDP marker system is suitable for determining polymorphism in the genotypes of the grapevine, as there was no monomorphic profile obtained for any variety. We observed that the amplicon patterns with all chosen primers were highly specific for all surveyed grapevine varieties. Chitinase expression analysis showed that it was highest in the Grűner Veltliner for both unmatured and matured grapes The most stable expression profiles were obtained for Weisser Riesling and Welschriesling when comparing unmatured and matured grapes. In Cabernet Sauvignon, the expression of chitinase was the most different among all the analysed varieties.

## Figures and Tables

**Figure 1 plants-11-03375-f001:**
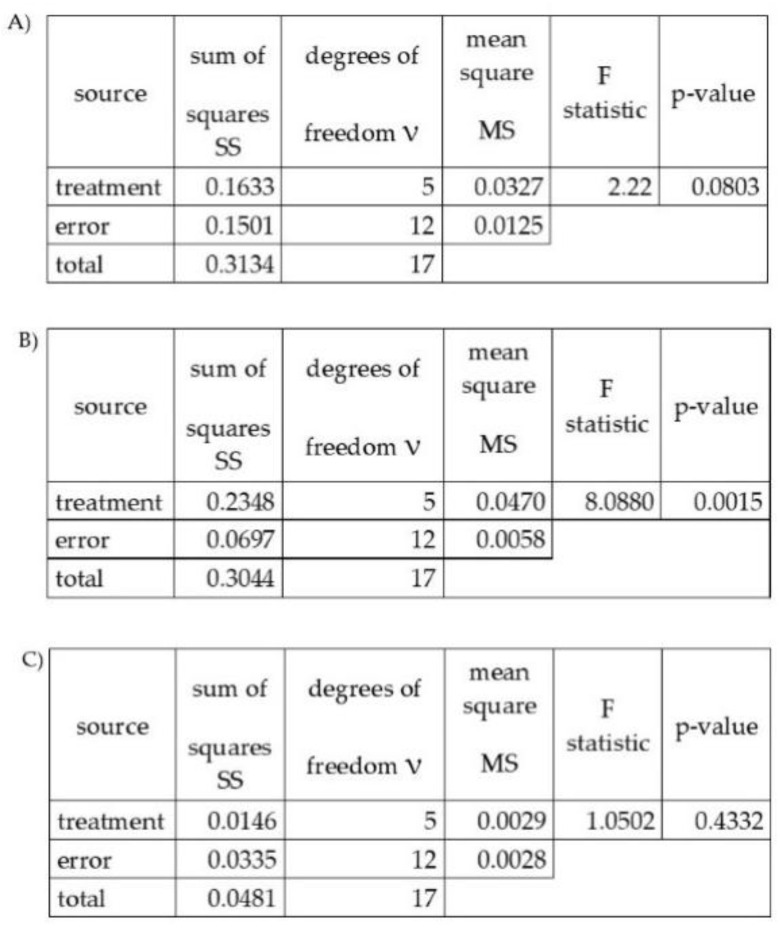
Results of ANOVA for individual microorganism parameters tested in the study. (**A**)—total viable count, (**B**)—coliform bacteria, (**C**)—microscopic filamentous fungi.

**Figure 2 plants-11-03375-f002:**
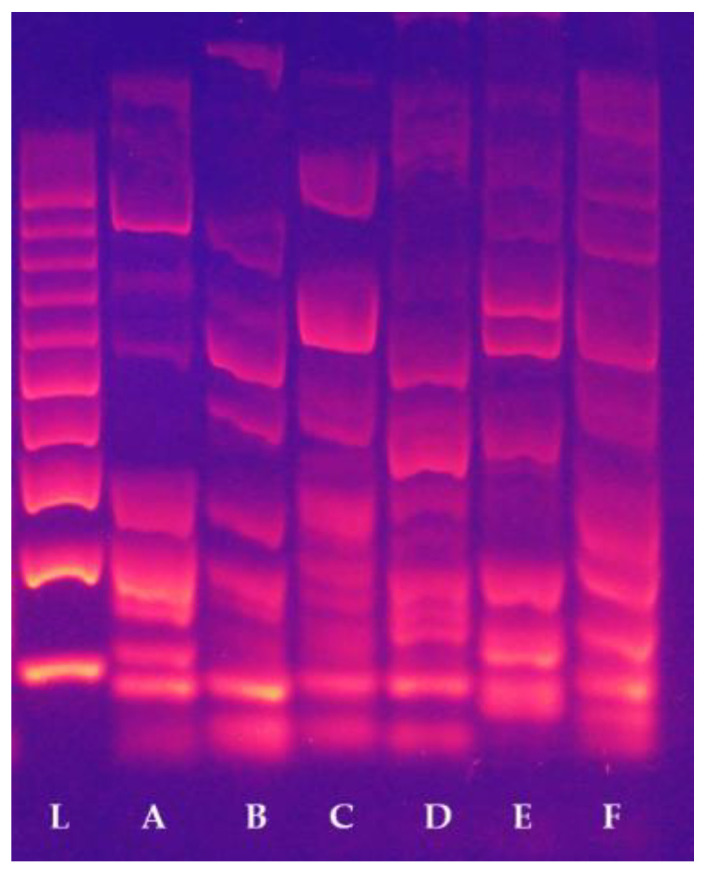
CDDP fingerprint pattern obtained by primer combination WRKY F1/R3. L—100 bp ladder; A—Grűner Veltliner; B—Weisser Riesling; C—Welschriesling; D—Chardonnay; E—Gewurztraminer; F—Cabernet Sauvignon.

**Figure 3 plants-11-03375-f003:**
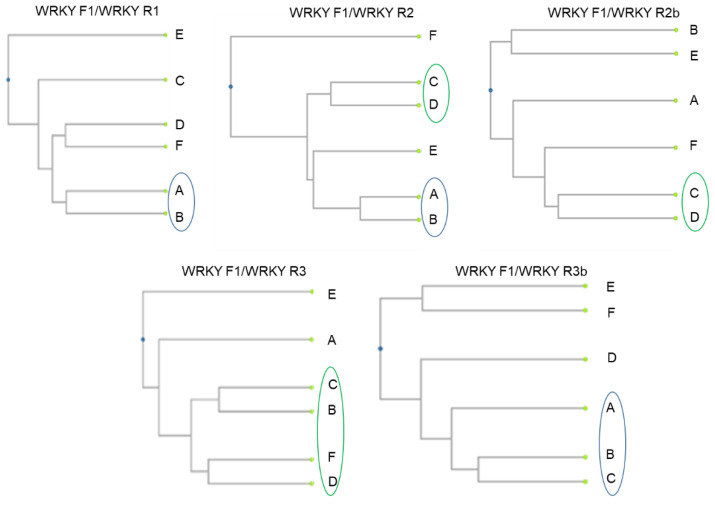
Dendrograms of Jaccard genetic similarity values among analysed *Vitis vinifera* L. genotypes for CDDP generated fingerprints. A—Grűner Veltliner; B—Weisser Riesling; C—Welschriesling; D—Chardonnay; E—Gewurztraminer; F—Cabernet Sauvignon; WRKY F1, WRKY R1-R3b—types of markers used in the conserved DNA-derived polymorphism [20].

**Figure 4 plants-11-03375-f004:**
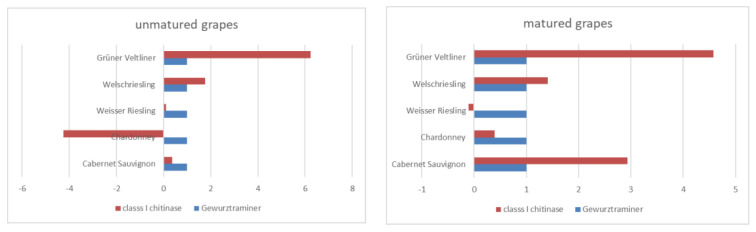
Comparison of expression profiles of unmatured and matured grapevine varieties analysed in the study.

**Table 1 plants-11-03375-t001:** Number of different group of microorganisms in log cfu/g.

Variety	Microorganisms		
	TVC	CB	MFY
Cabernet Sauvignon	3.19 ± 0.06 ^a^	1.19 ± 0.04 ^a^	1.14 ± 0.02
Chardonnay	3.32 ± 0.11	1.26 ± 0.06 ^b^	1.14 ± 0.05
Weisser Riesling	3.39 ± 0.15	1.47 ± 0.03 ^a,b^	1.18 ± 0.06
Welschriesling	3.34 ± 0.10	1.47 ± 0.07 ^a,b^	1.20 ± 0.05
Grűner Veltliner	3.51 ± 0.10 ^a^	1.32 ± 0.09	1.20 ± 0.05
Gewurztraminer	3.33 ± 0.11	1.46 ± 0.11 ^a^	1.21 ± 0.06

Level of confidence *p* < 0.05. Different superscript letters within the same column differ significantly. TVC—total viable count; CB—coliform bacteria; MFY—microscopic filamentous fungi.

**Table 2 plants-11-03375-t002:** Antimicrobial activity of different grape variety in mm.

Variety	Antimicrobial Activity in mm								
	SE	YE	PA	SA	EF	LM	CA	CK	CT
Cabernet Sauvignon	20.3 ± 0,57 ^a^	18.3 ± 0,57 ^a^	18 ± 1	15.6 ± 0.57	14.3 ± 0.57 ^a^	12.6 ± 0.7 ^a^	8.3 ± 0.57 ^a^	10.3 ± 0.57 ^a^	8.6 ± 0.57
Chardonnay	18.6 ± 0.57 ^a,b^	20.7 ± 0.57 ^a,b^	17.3 ± 0.57 ^b^	14.3 ± 0.57 ^b^	12.3 ± 0.57 ^a,b^	11.3 ± 0.57 ^b^	9.3 ± 0.57 ^b^	8.6 ± 0.57 ^a,b^	7.3 ± 0.57 ^b^
Weisser Riesling	21.3 ± 0.57 ^c^	18.3 ± 0.57 ^b,c^	17.6 ± 0.57	15.3 ± 0.57	14.6 ± 0.57 ^b,c^	12.3 ± 0.57 ^c^	7.6 ± 0.57 ^b,c^	6.3 ± 0.57 ^a,b,c^	7.3 ± 0.57 ^c^
Welschriesling	18.3 ± 0.57 ^a,c,d^	18.6 ± 0.57 ^d^	17.6 ± 0.57	14.6 ± 0.57 ^d^	14.3 ± 0.57 ^b,d^	14.3 ± 0.57 ^a,b,c,d^	9.3 ± 0.57 ^c,d^	7.6 ± 0.57 ^a,d^	7.6 ± 0.57
Grűner Veltliner	19.3 ± 0.57 ^c,e^	18.6 ± 0.57 ^e^	18.3 ± 0.57	15.3 ± 0.57	13.6 ± 0.57 ^b^	15.3 ± 0.57 ^a,b,c^	8.3 ± 0.57 ^b,e^	8.6 ± 0.57 ^a,c,e^	8.3 ± 0.57
Gewurztraminer	22.3 ± 0.57 ^a,b,d,e^	21.6 ± 0.57 ^a,c,d,e^	19.3 ± 0.57 ^b^	16.3 ± 0.57 ^b,d^	14.3 ± 0.57 ^a,b,c,d^	16.6 ± 0.57 ^a,b,c,d^	11.3 ± 0.57 ^a,c,d,e^	10.6 ± 0.57 ^b,c,d,e^	9.3 ± 0.57 ^b,c^

Level of confidence *p* < 0.05. Different superscript letters within the same column differ significantly. SE—*Salmonella enteritidis*; YE—*Yersinia enterocolitica*; PA—*Pseudomonas aerogonosa*; SA—*Staphylococcus aureus*; EF—*Enterococcus faecalis*; LM—*Listeria monocytogenes*; CA—*Candida albicans*; CK—*Candida kruseii*; CT—*Candida tropicalis*.

**Table 3 plants-11-03375-t003:** Characteristics of Jaccard coefficients of similarity values and polymorphism of individual primer pairs used in the study.

Primer Combination	Min JCS	Max JCS	Average JCS	Polymorphism
WRKY F1/R1	0.148 (A–E; D–E)	0.481 (A–B)	0.324	97%
WRKY F1/R2	0.200 (C–F)	0.778 (A–B)	0.595	90%
WRKY F1/R2b	0.143 (D–F)	0.634 (B–C)	0.405	91%
WRKY F1/R3	0.357 (C–E)	0.667 (B–C)	0.638	95%
WRKY F1/R3b	0.267 (A–E, D–E)	0.583 (C–D)	0.406	94%

**Table 4 plants-11-03375-t004:** Sequences of primers used in the study [20].

Primer Name	Primer Characteristics		
	Sequence 5′-3′	% GC	Length
WRKY-F1	TGGCGSAAGTACGGCCAG	67	18
WRKY-R1	GTGGTTGTGCTTGCC	60	15
WRKY-R2	GCCCTCGTASGTSGT	67	15
WRKY-R2b	TGSTGSATGCTCCCG	67	15
WRKY-R3	GCASGTGTGCTCGCC	73	15
WRKY-R3b	CCGCTCGTGTGSACG	73	15

## Data Availability

Not applicable.

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
