# Peer review of "Molecular Fingerprinting and Microbiological Characterisation of Selected Vitis vinifera L. Varieties"

_plants, 2022, doi:10.3390/plants11233375_

Round 1

Reviewer 1 Report

Dear Author,

After peer review of the MS entitled “Molecular fingerprinting and microbiological characterization of selected Vitis vinifera L. varieties”, I suggest the MS needs major revision and cannot be accepted in its present form. The MS may be revised in light of the comments below:

1.      The abstract should start with a sentence of rationale.

2.      Abbreviation should be avoided in the abstract without prior mention of the full form. The mention of full form is required prior to mention of their abbreviated forms throughout the MS.

3.      In abstract correct the number of varieties mentioned.

4.      The starting line of the introduction section is not appropriate. Redraft it.

5.      In line 126 correct the table number.

6.      The sentence in line no. 119-122 is not well written and needs to be redrafted.

7.      In table 1 and 2 what does ‘a, b, c, d, e’ indicate needs to be mentioned.

8.      In figure 1 put the abbreviations used as foot note.

9.      The polymorphism obtained needs to be mentioned.

10.  The result obtained for each of the primers needs to be provided as well as mention the best performing primer.

11.  Each of the subsection under result and discussion should end with a concluding remark from the author.

12.  In line 309, it should be 1.5 % instead of 1,5 %. Correct it.

13.  The sequence for the primers used needs to be provided.

14.  The result of the ANOVA performed needs to be mentioned as well.

15.  The English language of the MS needs to be improved by a native speaker.

16.  The future prospects needs to be mentioned.

Author Response

Reviewer 1

The abstract should start with a sentence of rationale.

Was corrected.

Abbreviation should be avoided in the abstract without prior mention of the full form. The mention of full form is required prior to mention of their abbreviated forms throughout the MS.

Was checked in manuscript and rewritten.

The starting line of the introduction section is not appropriate. Redraft it.

Was corrected.

The sentence in line no. 119-122 is not well written and needs to be redrafted.

Was rewritten and arranged more properly.

In table 1 and 2 what does ‘a, b, c, d, e’ indicate needs to be mentioned.

Was corrected.

In figure 1 put the abbreviations used as foot note.

Was corrected.

The polymorphism obtained needs to be mentioned. The result obtained for each of the primers needs to be provided as well as mention the best performing primer.

Was added.

Each of the subsection under result and discussion should end with a concluding remark from the author.

Was added.

The sequence for the primers used needs to be provided.

Was added.

The result of the ANOVA performed needs to be mentioned as well.

Was added.

Reviewer 2 Report

The manuscript " Molecular fingerprinting and microbiological characterization of selected Vitis vinifera L. varieties " falls within the scope of the journal and constitutes a work with relevant information and results on biological control. This paper presents a considerable amount of work but has a number of shortcomings. In summary, with respect to the molecular characterization of different grape and their antibacterial activities in the study, from my opinion the authors should add the gels profile and table contain data revealed from molecular markers. Also, the figures showed its antimicrobial effects against different pathogens.  . Finally, the English should be carefully reviewed since the text contains a number of typos and structural problems. you can see my suggestions in the attachment.

Author Response

Reviewer 2

The discussion part relating to the presented results should be in paragraph.

Was rearranged.

Introduction - more information regarding WRKY selection should be presented.

Was added.

Figure 2 - lable should be expanded to properly refer to the presented data

Was added.

Reviewer 3 Report

Major comments:

The discussion part relating to the presented results should be in paragraph.

In general, presented results should be discussed more thoroughly.

Introduction - more information regarding WRKY selection should be presented.

Figure 2 - lable should be expanded to properly refer to the presented data

In general, manuscript needs to be properly aranged into section (introduction, methods, results and discussion). 

Minor:

Please improve overall quality of the manuscript, language, phrasing, and structure; some examples:

CDDP - please unravel abbreviation. 

L17 - Vitis vinifera - italics

L20 - WRKY - italics

L20 - 'all used primers" - rephrase 

L25 - should such precise values to be presented in the abstract with SD?

L40 - 'defence to' rephrase

L39-L48 - limit usage of these/this as it counfese the reader

L52 - remove comma before too

L78 'in our study according to" rephrase

Section 2.1 - whole first part sounds more like introduction than results L111-L115 also, while line L116-L121 more like methods

Author Response

Reviewer 3

Missing letters of significancy in tables.

Was corrected.

Adding gels and primer pairs relevant informations.

Was added.

Round 2

Reviewer 1 Report

Dear Authors 

Although MS has significant revised some minor references checking required  the quality is good for acceptance 

Author Response

Thank you for your time and improvement suggestions. Manuscript as well as references were checked again.

Reviewer 2 Report

Dear authors

Many thanks for authors, all comments were revised and done

Author Response

Thank you for your time and improvement suggestions.

Reviewer 3 Report

The manuscript has been improved. 

Author Response

(The authors gave the same response as above.)
